# Reoperation for Recurrent and Persistent Cushing’s Disease without Visible MRI Findings

**DOI:** 10.3390/jcm11226848

**Published:** 2022-11-20

**Authors:** Baofeng Wang, Shuying Zheng, Jie Ren, Zhihong Zhong, Hong Jiang, Qingfang Sun, Tingwei Su, Weiqing Wang, Yuhao Sun, Liuguan Bian

**Affiliations:** 1Department of Neurosurgery, Ruijin Hospital, Shanghai Jiao Tong University School of Medicine, Shanghai 200025, China; 2Department of Endocrinology and Metabolic Disease, Ruijin Hospital, Shanghai Jiao Tong University School of Medicine, Shanghai 200025, China

**Keywords:** Cushing’s disease, MRI, persistence, recurrence, repeat surgery

## Abstract

Purpose: Transsphenoidal surgery is the first-line treatment for Cushing’s disease (CD), even with negative preoperative magnetic resonance imaging (MRI) results. Some patients with persistent or recurring hypercortisolism have negative MRI findings after the initial surgery. We aimed to analyze the efficacy of repeat surgery in two groups of patients and determine if there is an association between positive MRI findings and early remission. Patients and Methods: Clinical, imaging, and biochemical information of 42 patients who underwent repeat surgery by a single neurosurgeon between 2002 and 2021 was retrospectively analyzed. We compared the endocrinological, histopathological, and surgical outcomes before and after repeat surgery among 14 CD patients with negative MRI findings and 28 patients with positive MRI findings. Results: Immediate remission was achieved in 29 patients (69.0%) who underwent repeat surgery. Among all patients, 28 (66.7%) had MRI findings consistent with solid lesions. There was no significant difference in remission rates between the recurrence and persistence groups (77.8% vs. 57.1%, odds ratio = 2.625, 95% confidence interval = 0.651 to 10.586). Patients in remission after repeat surgery were not associated with positive MRI findings (odds ratio = 3.667, 95% confidence interval = 0.920 to 14.622). Conclusions: In terms of recurrence, repeat surgery in patients with either positive or negative MRI findings showed reasonable remission rates. For persistent disease with positive MRI findings, repeat surgery is still an option; however, more solid evidence is needed to determine if negative MRI findings are predictors for failed reoperations for persistent hypercortisolism.

## 1. Introduction

Transsphenoidal pituitary surgery is the primary treatment choice for patients with Cushing’s Disease (CD), which has a reported remission rate of 70% to 90% [1,2]. However, hypercortisolism persists in some of these surgical patients and recurs in 3–29% of patients, even in those who have benefited from remission for more than a decade [3,4].

In cases in which the primary surgery failed, serval treatments are considered, including reoperation, medication, conventional radiotherapy, radiosurgery, and bilateral adrenalectomy [4]. With remission rates as high as 87% [5], reoperation is a feasible option worth considering. Although some studies have concentrated on the risk factors and long-term outcomes of repeated transsphenoidal surgery [6,7], the necessity of reoperation in patients with varied clinical, imaging, and pathological characteristics has not been adequately discussed. Reoperation is considered when lesions remain visible on magnetic resonance imaging (MRI), given that tumor removal will likely lead to remission, even if it is located in the cavernous sinus [8]. Nevertheless, the incidence of positive MRI findings is typically low in CD patients with either recurrent or persistent disease [5,9,10,11]. Furthermore, MRI has limitations in revealing the accurate structures of the operated area due to distorted anatomy related to the formation of granulation tissue and inflammatory changes after the initial surgery [12]. Unlike the considerable remission rate achieved after the first operation despite negative MRI findings [1], the decision to perform a second operation without visible lesions detected on MRI is challenging for neurosurgeons. These uncertainties emphasize the importance of discussing the risk factors and the necessity of repeat surgery, especially for patients with negative radiological results.

Our retrospective study aimed to ascertain the treatment preference for reoperation in patients with persistent and recurrent CD and evaluate the significance of MRI findings for selecting patients that are likely to benefit from reoperation. Furthermore, we aimed to provide a reference for surgeons in making decisions on repeat surgical intervention for patients who are most likely to benefit, thereby improving the remission rates associated with reoperation.

## 2. Patients and Methods

We retrospectively identified patients with CD treated with repeated transsphenoidal surgery between 2002 and 2021 at our institution. Patients with three or more pituitary surgeries were excluded from the present study. The preoperative and postoperative evaluations of the first surgeries are shown in Table 1. All patients fulfilled the following inclusion criteria: persistent hypercortisolism after initial surgery or recurrence after remission with a period of normocortisolism or adrenal insufficiency.

This study included 42 patients aged 44.4 ± 14.6 years at the time of the repeat operation (Appendix A). The median interval between the two operations was 43 months (interquartile range [IQR] = 18–90). The median follow-up duration after the second operation was 15.5 months (IQR = 4–59).

### 2.1. Diagnosis

The diagnostic criteria for recurrence in the present study included new onset or recurrence of symptoms, clinical features, serum cortisol level, 24 h urinary-free cortisol (UFC) level, and biochemical tests (low-dose dexamethasone suppression test and high-dose dexamethasone suppression test (HDDST)), which are frequently used to define CD remission, recurrence, and persistence. An algorithm that is currently used in biochemical assessment and management of recurrent and persistent disease is shown in Figure 1. All tests were performed in a College of American Pathologists-accredited laboratory (No. 7217913). Serum cortisol and UFC were examined using an Access Immunoassay System (Beckman Coulter Inc., Fullerton, CA, USA). The normal ranges were 6.7–22.6 µg/dL and 21–111 µg/24 h, respectively. Plasma adrenocorticotropic hormone (ACTH) levels were measured using an ELSA-ACTH immunoradiometric method (Cisbio Bioassays, Codolet, France). The normal range was 12–78 pg/mL. A serum cortisol value of less than 5 μg/dL was considered to indicate remission. Patients who were not considered to be in remission were discharged and routinely evaluated 6 months after surgery for possible delayed remission. Patients were administered oral cortisone and gradually withdrawn to a physiologic replacement dose after 1 month. The yearly follow-up visit included physical examinations and serum cortisol, UFC, and plasma ACTH assessments. MRI was not performed routinely after surgery unless persistent or recurrent hypercortisolism was confirmed biochemically, as postoperative imaging may not be reliably interpreted for hormone-active pituitary adenoma.

Contrast-enhanced pituitary MRI at our center was conducted to facilitate diagnosis and surgical planning using a superconducting magnet 1.5/3.0 Tesla scanner (SIGNA; GE Healthcare, Chicago, IL, USA). Before gadolinium injection (0.01 mmol/kg gadopentetate dimeglumine; Magnevist, Berlex Laboratories, Inc., Montville, NJ, USA), T1-weighted spin echo and T2-weighted turbo spin echo images were obtained in the coronal and sagittal planes. Beginning simultaneously with gadolinium injection, coronal and sagittal T1-weighted spin echo images were obtained 2 min after the injection. Imaging studies were independently reviewed by a neuroradiologist, endocrinologist, and the patient’s neurosurgeon. Pituitary imaging prior to the first surgery performed outside of our center was acquired and re-interpreted by the same team. Full agreement was reached on the positive nature of the MRI findings. Otherwise, when MRI findings appeared normal or interpretation was ambiguous, the MRI findings were considered negative.

Meanwhile, bilateral inferior petrosal sinus sampling (BIPSS) with or without vasopressin (available after 2015) stimulation was performed in nine patients who experienced recurrence but lacked initially positive ACTH staining on the first histological examination to reconfirm whether the Cushing’s syndrome diagnosis was pituitary-dependent. Two patients were evaluated by BIPSS, although the initial pathology was positive. Regarding persistent disease, among eight patients without positive ACTH staining in their first pathological assessment, five were confirmed by positive BIPSS results and five were confirmed by visible radiological lesions. Only one patient with negative ACTH-staining adenoma underwent repeat surgery with either negative BIPSS results or negative imaging findings.

### 2.2. Surgical Procedure

The same surgeon performed surgery on all patients via the mononostril transsphenoidal approach under a microscope or endoscope (available from December 2015). The initial location prior to the first operation did not guide the resection during repeat surgery. For each patient with positive MRI results, the imaging-identified areas for adenoma were biopsied as frozen sections for the initial pathological evaluation. Subsequent resection with a rim of pituitary tissue around the tumor cavity was conducted to confirm neoplasm-free margins. No further exploration was performed before frozen pathology confirmation was available unless the BIPSS result showed an increased ACTH level on the other side.

For invisible tumors on MRI, the dura mater was opened widely to facilitate exploration of the whole gland, starting from the initial location on MRI before the first surgery or the side with the higher ACTH level in the BIPSS, if available. If no obvious tumor was identified on this side by the neurosurgeon intraoperatively, half of the gland was resected using the guidance of BIPSS lateralization.

If a tumor was frozen pathologically and identified after half of the gland was removed, the residual gland remained unresected and was only gently explored and sampled in the most suspected area. In some circumstances in which the frozen section was negative, it was subjected to a subtotal adenohypophysectomy of the intermediate lobe and neurohypophysis.

If invasive adenoma characteristics were also identified, the involved dura and medial wall of the cavernous sinus were resected or coagulated. A sample was collected for postoperative pathological confirmation, if available.

### 2.3. Outcome

Patients were defined as being in remission with an immediate postoperative serum cortisol nadir <5 μg/dL or 24 h UFC at a normal level [13]. Persistent hypercortisolism was defined as an increased postoperative UFC level, while recurrent hypercortisolism was defined as a reappearance of hypercortisolism after a period of normocortisolism or adrenal insufficiency.

### 2.4. Statistical Analysis

Descriptive statistics are presented as means ± standard deviations when normally distributed or medians and ranges when not normally distributed to describe patient outcome measures and incidence of remission among the study population. Statistical significance was set at a *p* value < 0.05. Fisher’s exact test was used to compare proportions of categorical measures between groups. All analyses were conducted using Instat (GraphPad Software, San Diego, CA, USA).

## 3. Results

### 3.1. Patient Characteristics

The basic information and perioperative evaluations of the two operations are shown in Table 1 and Appendix A. Among all 27 recurrent cases, the preoperative MRI before the first operation showed a definite pituitary adenoma. The other 12 patients with persistent hypercortisolism had positive MRI findings before the first surgery. The remaining three patients with negative radiographic findings were diagnosed with CD and underwent the first transsphenoidal surgery (TSS) based on their endocrinological results.

For patients with confirmed persistent or recurrent CD, the imaging findings prior to the second operation of 14 individuals were negative (no solid evidence of tumors), and 28 clearly had positive results for the presence of a solid lesion. All patients who underwent a second surgery for recurrent or persistent hypercortisolism after the initial surgery were endocrinologically re-evaluated before the repeat surgery. There were 38 cases with positive HDDST results among 42 patients. BIPSS was performed in 18 patients with only one that did not reach the criteria of pituitary origin.

### 3.2. Outcome

In our study, 29 of 42 patients (69.0%, 22 recurrent and 7 persistent cases of CD) were in remission after the repeat operation without additional therapy during follow-up (Appendix A). At follow-up, compared with patients with persistent disease, the recurrence group had a higher remission rate, although the difference was not significant (77.8% [21/27] vs. 57.1% [8/15]; *p* > 0.05; odds ratio = 2.625, 95% confidence interval = 0.651 to 10.586). Negative preoperative MRI findings were not associated with lower odds of immediate remission after repeat surgery (*p* > 0.05; odds ratio = 3.667, 95% confidence interval = 0.920 to 14.622; Table 2).

### 3.3. Association between Outcomes and MRI Findings

The remission rates of the persistent and recurrent disease groups with positive and negative MRI findings prior to the second procedure are shown in Table 2. Twenty-nine patients whose MRI findings revealed the existence of pituitary adenomas achieved successful outcomes after reoperation (Representative case, #19, Figure 2). The other seven patients who experienced recurrent or persistent hypercortisolism without clear imaging evidence of tumor appearance also benefited from reoperation (Representative case, #11, Figure 3).

### 3.4. Pathology

Respectively, 15/27 (55.6%) and 7/15 (46.7%) patients with recurrent and persistent hypercortisolism had ACTH-positive staining in the first pathological findings. Among patients who achieved remission after the second operation, 20 of 29 patients had confirmed adenoma with positive ACTH pathological staining, while 3 patients with adenoma were ACTH-negative. There were five patients that did not achieve remission even though they had positive ACTH-staining adenoma in the second pathological examination. Meanwhile, five patients achieved remission, although no adenomas were found in their pathological specimens. Overall, positive pathology after either the initial or repeated surgery was not a significant predictor for remission after the second surgery.

### 3.5. Complications

Four of forty-two patients experienced major postoperative complications and underwent medical or surgical interventions. Most patients recovered well after the second operation, except in one case with persistent hypercortisolism, where a severe intracranial infection led to death. Another three cases with cerebral spinal fluid leakage related to the second operation were successfully surgically repaired afterwards.

Hypopituitarism was a common complication in this subgroup of CD. All of the patients in remission after the second TSS underwent glucocorticoid replacement therapy (hydrocortisone or cortisone), adjusted according to the 24 h UFC. A total of 20 patients (20/29, 68.9%) underwent thyroxine replacement therapy. Three patients (3/29, 10.3%) had permanent diabetes insipidus. In the non-remission group, five patients (5/13, 38.5%) experienced hypothyroidism, and two patients (2/13, 15.4%) had permanent diabetes insipidus.

## 4. Discussion

In the present study, we reported outcomes for 42 patients undergoing repeat TSS for recurrent and persistent disease in which an overall remission rate of 69.0% was achieved. Immediate remission rates after reoperation for recurrence have been reported in the literature up to 87% [13,14], which is similar to those of other second-line therapies such as radiation therapy and medical treatment. The CD recurrence rate after the initial TSS is reportedly 10–25% with a follow-up time of 10 years [15,16,17]. Ram et al. reported that surgeons performed a second TSS immediately after the first TSS when the postoperative serum cortisol level did not meet the standard level of remission. With an interval time of 1 to 6 weeks, 71% of patients with persistent disease achieved immediate remission, and 53% (9/17) achieved long-term remission [13]. Another study showed a remission rate of 70% with reoperation performed within 10 days [18]. A second TSS reportedly leads an additional 8% of patients to long-term CD remission [3]. Recurrence groups had slightly higher remission rates, which are insignificant when compared with persistent groups in the present study. Similar findings are demonstrated in the study by Ram et al. implicating that failure of the initial surgery suggested that the patient was more difficult to treat successfully with surgery than most patients with recurrence [13]. Therefore, the selection criteria for potential patients and reoperation strategies require further discussion.

### 4.1. Surgical Strategy

The surgical strategy for the initial CD surgery varies depending on the major concerns of different pituitary surgeons. Some surgeons intend to preserve more normal gland tissue during surgery while others chase higher remission rates. Selective adenectomy is a reasonable choice for visible tumors. Several authors adopted a slightly extended resection with a rim or sometimes 2–3 mm of like-normal tissue around the tumor, which could be considered a partial hypophysectomy [19,20]. A hemi-hypophysectomy is more common in cases in which no tumor was identified during the operation, and the MRI or BIPSS results indicated remarkable lateralization of the tumor origin [21]. Wide exploration of the contralateral side should also be conducted in cases in which BIPSS results are inconsistent with the MRI findings, which may help identify tiny tumors. More extensive procedures, including subtotal or sometimes total pituitary gland resection, have been performed to maximize remission rates up to 75.9–81.8% [20,22], which may be a reasonable recommendation when imaging/intraoperative findings are not definitive, considering the negative impacts on reoperated patients with persistent hypercortisolism rather than hypopituitarism. Interestingly, pathological confirmation rates are fairly low in cases with extended resection even though they show high remission rates. There seems to be a current trend of surgeons performing a partial hypophysectomy, as a total hypophysectomy can lead to hypopituitarism [5,22,23], given that it may not obviously increase remission rates and may decrease quality of life [24].

### 4.2. MRI Findings

Regarding radiological findings, we emphasize that negative MRI findings do not necessarily indicate the inexistence of pituitary adenomas or negative pathological results. A number of cases in the study by Wagenmakers et al. showed that remission achieved after repeated transsphenoidal surgery was not predictable by positive MRI findings before the first or second operation [10]. Preoperative MRI provides a reference for the diagnosis of pituitary adenomas, although it has a limited predictive function for patient prognosis [9], especially for the repeat operation in which the original anatomical structure was more or less destroyed in the initial surgery. A positive MRI finding before the second operation should promote confidence in surgeons. The remission rate after reoperation with positive MRI findings was reportedly as high as 72.7% [10]. According to our study, the two positive-MRI groups with different initial surgical outcomes showed higher remission rates, albeit insignificantly. Positive MRI findings suggest better endocrinological outcomes may be achieved by a second operation in both recurrent and persistent disease groups compared with patients with negative imaging findings. An excellent remission rate (more than 80%) was achieved in the recurrent group with positive MRI findings, thus encouraging a repeat TSS. An acceptable remission rate (over 60%) close to those of alternative treatment options was observed in the recurrent group with negative MRI findings, as well as the persistent group with positive MRI findings. We noted that one patient with persistent CD and negative MRI findings achieved remission after reoperation. Therefore, whether a second surgical treatment is beneficial for these patients should be carefully considered.

Regarding the recurrent or persistent cases of CD, patients underwent an initial surgery, and we regarded the MRI findings as a possible method to assist in decision making. A second operation is considered when visible lesions remain on MRI under the assumption that removal of the residual tumor leads to remission of the disease. Meanwhile, some recurrent and persistent patients with negative MRI findings also benefited from reoperation. Furthermore, MRI has its limitations in revealing the accurate structures of the originally operated area. The distortion and cicatrization from the previous operation and material packing in the sellar region lead to confusion [12,25]. Unlike the considerable remission rate achieved after the initial operation despite negative MRI findings, reoperation without certain lesion detection on MRI is associated with dissatisfactory remission rates [1], similar to the results of our study. Nevertheless, Knappe and Lüdecke [9] presented a different opinion regarding the significance of MRI findings and reported that it was not usually helpful for determining therapeutic strategies due to its low incidence of detecting existing microadenomas (missed diagnosis in 38–70% of cases). However, the BIPSS results in these cases in which MRI revealed no definitive information on tumors are therefore critical for surgeons to ascertain the pituitary origin of the disease, although another study suggested that MRI and BIPSS do not help locate recurrent tumors [10]. MRI may not help identify tumors in the cavernous sinus or other parasellar regions.

### 4.3. Pathology

We compared the pathological results and remission situations of recurrent patients and persistent patients and failed to find any relationship between pathological results and remission expectations. These findings are supported by the findings of Ram et al. [13], in which no tumors were found in 11 of 17 patients during the second procedure, and 6 of 11 patients achieved remission. In a series by Locatelli et al. [11], no tumors were found in 8 of 12 patients during the second operation, and 5 had surgical remissions. Even in cases of remission, the positive rate of pathological exams was not as high as expected. There was no significant difference in remission rates between patients grouped by pathological results or one-to-one correspondence between histopathological confirmation and surgical outcomes [11]. To date, little evidence supports the prediction of reoperation outcomes by either of the two pathology results.

### 4.4. Other Considerations and Factors

In patients with recurrent and persistent hypercortisolism after their first operation, it was difficult to identify solid lesions on MRI compared with the initial preoperative scans. Notably, BIPSS may provide more information, especially for patients who did not undergo this test before the first operation. Moreover, it may help avoid unnecessary repeat TSS in patients with persistent hypercortisolism by revealing false positives for pituitary ACTH overproduction. BIPSS results have the potential to not only confirm the pituitary origin of the condition (despite the fact that the first histological examination did not show ACTH-positive staining) but also to guide exploration and decision making for a hemi-hypophysectomy or accessing the cavernous sinus, especially for patients without obvious tumors identified intraoperatively. Careful dissection is highly recommended on the side of the obviously lateralized BIPSS results, which sometimes also indicate cavernous sinus invasion not shown on MRI and the necessity of opening the medical wall to achieve extended exploration. The predictive value of BIPSS lateralization in repeated surgery requires further investigation, although it is not optimal in native patients with CD [26].

According to a study by Lonser et al. [27], over 20% of CD patients had cavernous sinus invasion that was confirmed histologically. The authors advocated for complete resection, including the invaded sella dura and medial cavernous sinus wall by an experienced surgeon’s hands. Notably, endoscopy with magnification and lighting provides a panoramic view to facilitate extended exploration of the sella, including the cavernous sinus, compared with the microscope-based approach. Micko et al. demonstrated that an endoscope allows for a radical inspection of the entire medial wall of the cavernous sinus [28] and increases the lateral angle of visualizations to facilitate differentiation between tumor tissues and other tissues. These advantages over the microscopic transsphenoidal approach are critical for recurrent and unremitted cases; however, further studies with larger sample sizes are needed to verify this conclusion.

### 4.5. Other Adjunctive Treatments to Repeat Surgery

Previous studies have noted that ketoconazole may contribute to enhanced tumor appearance on MRI to facilitate pituitary resection in some circumstances [29]. Castinetti et al. reported that visible lesions may be identified on MRI in one-third of patients who were administered ketoconazole [30].

In the literature, reoperation for persistent cases without visible lesions on MRI is rarely satisfactory [31], although these patients may benefit from radiosurgery using the entire sellar region as the therapeutic target [32]. The hormonal normalization was achieved after radiosurgery in half of the cases, including those with negative MRI findings [33]. In general, the radiosurgery outcomes and the less commonly used radiotherapy are more favorable, particularly in MRI-negative cases with persistent hypercortisolism compared with repeat surgery, with potentially fewer complications and a shorter length of hospital stay [34,35]. Salvage TSS for refractory CD after radiation therapy has rarely been reported [36] owing to the difficulty of disrupting surgical landmarks, the formation of scar tissue, and the effects of preoperative radiotherapy [34].

Bilateral adrenalectomy is generally considered the ultima ratio in patients who fail to respond to other treatment options. However, patients who undergo bilateral adrenalectomy will require lifelong surveillance of the corticotroph tumor’s progression, which may lead to Nelson’s syndrome, via MRI and ACTH measurements. Most experts agree that selective transsphenoidal adenomectomy should be recommended as the first-line therapy in patients with Nelson’s syndrome before extrasellar expansion of the tumor occurs [37].

### 4.6. Limitations

Similar to previous studies, our sample size was not large enough to conduct powerful statistical analyses. Some patients lost during follow-up limited the evaluation of long-term outcomes in the current study. We observed a trend in the predictable values of positive preoperative MRI findings, which is not enough evidence to support an apparent relationship. A potential weakness of the present study is that the outcome was only focused on the biochemical benefits of remission after surgical intervention, possibly leading to an underestimation of the risks of hypopituitarism and decreased quality of life. Indeed, larger case series are needed to further investigate the potential predictive factors and best surgical strategy.

## 5. Conclusions

Patients with initial surgical treatment may experience hypercortisolism without positive MRI findings in both recurrent and persistent disease. Our findings suggest that for most patients who experience recurrent or persistent CD, reoperation should be an option even with negative MRI findings. However, further comprehensive investigation on recurrent or persistent CD patients is required. Larger groups of surgically treated CD patients with long follow-up periods should be evaluated to improve reoperation outcomes and determine the appropriate selection criteria for repeat surgery, especially for persistent CD patients.

## Figures and Tables

**Figure 1 jcm-11-06848-f001:**
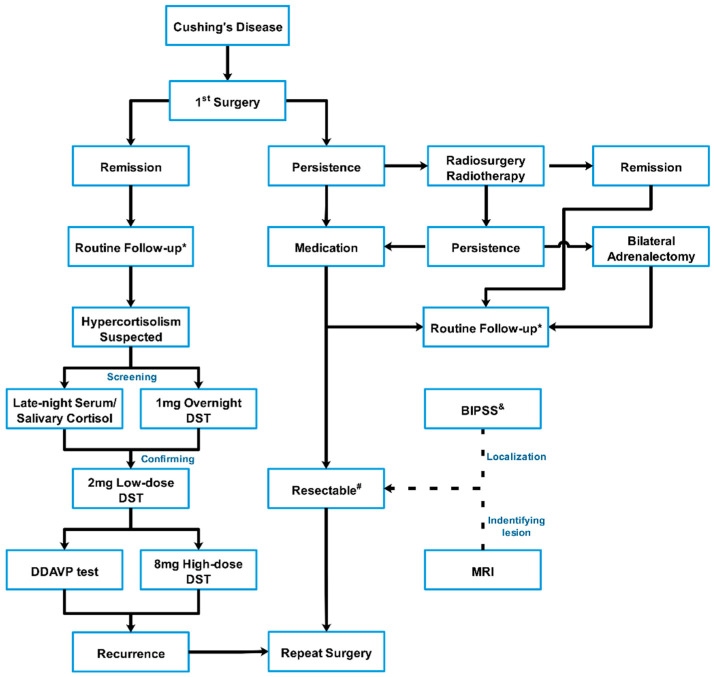
Algorithm of the biochemical assessment and treatment of persistent and recurrent Cushing’s disease.

**Figure 2 jcm-11-06848-f002:**
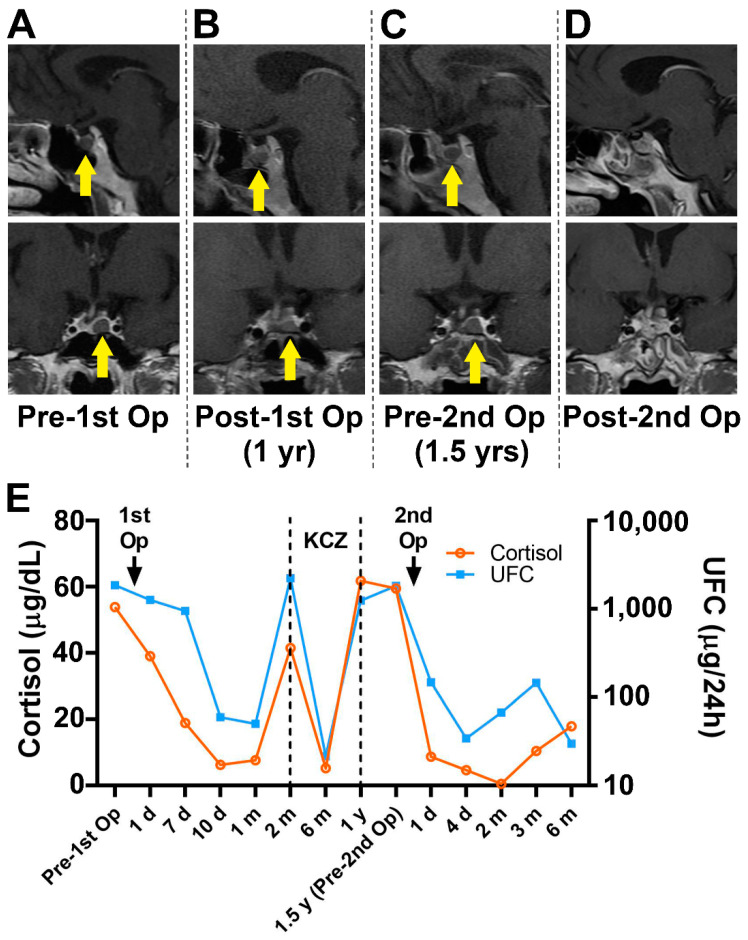
Preoperative and postoperative MR images of the two operations (**A**–**D**) demonstrate an in situ relapsed intrasellar mass (yellow arrow). Biochemical results obtained before and after the operations (**E**) show the tumor-related hormone change. KCZ, ketoconazole; MR, magnetic resonance.

**Figure 3 jcm-11-06848-f003:**
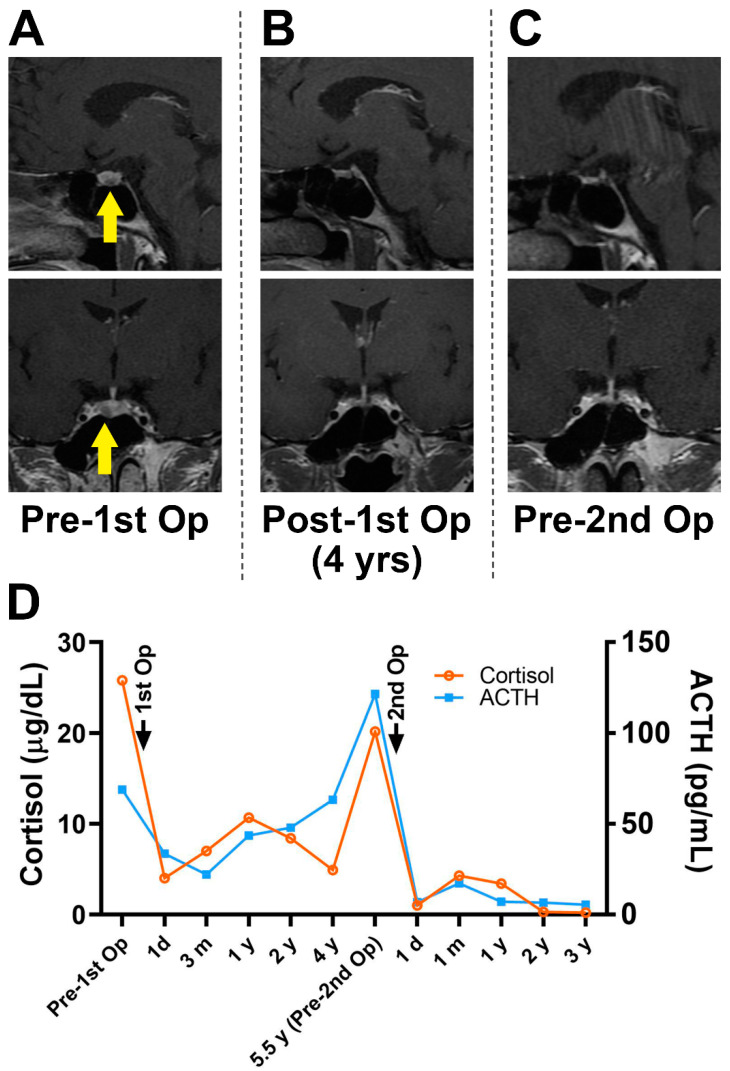
MR images (**A**) demonstrated a pituitary microadenoma on the left side (yellow arrow) before the first operation but not at the subsequent follow-ups (**B**,**C**). The biochemical results obtained before the second operation (**D**) revealed hypercortisolism indicating relapse without obvious MRI confirmation. MR, magnetic resonance; MRI, magnetic resonance imaging.

**Table 1 jcm-11-06848-t001:** Preoperative characteristics of the initial surgery.

	Recurrence Group	Persistence Group
*Age (Mean ± SD)*	41.7 *± 12.3*	33.4 *± 10.6*
*Gender*		
Male	3	6
Female	24	9
*MRI*		
Visible lesion	27	12
Negative	0	3
*HDDST*		
+	14	10
−	1	1
NA/NP	12	4
*BIPSS*		
+	5	2
−	3	0
NA/NP	19	13
*The surgeon of the initial procedure*		
Same	15	7
Different	12	8

Age, age at the initial operation; MRI negative, no tumor identified; HDDST−, failure of suppression after dexamethasone administration; HDDST +, both of serum cortisol and 24 h UFC suppressed after dexamethasone administration; BIPSS+, central to peripheral ACTH gradients ≥ 2 without CRH/DDAVP stimulation; BIPSS−, central to peripheral ACTH gradients < 2 without CRH/DDAVP stimulation; NA, no available record; NP, not performed.

**Table 2 jcm-11-06848-t002:** The remission rate of the recurrent and persistent hypercortisolism patients with or without positive MRI findings.

	Recurrence Group		Persistence Group		Combined	
	MRI +	MRI −	MRI +	MRI −	MRI +	MRI −
Remission	15	6	7	1	22	7
Non-Remission	2	4	4	3	6	7
Remission Rate	88.2%	60.0%	63.6%	25.0%	78.6%	50.0%

## Data Availability

All data generated or analyzed during this study are included in this article. Further enquiries may be directed to the corresponding authors.

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
