# Peer review of "Reoperation for Recurrent and Persistent Cushing’s Disease without Visible MRI Findings"

_jcm, 2022, doi:10.3390/jcm11226848_

Round 1

Reviewer 1 Report

The work is original and very interesting for the readers. It is only suggested to shorten the discussion, since it is very long.

Author Response

Thank you for your positive comments. The treatment preference for patients with persistent or recurring hypercortisolism have negative MRI finding after initial surgery is still a challenge for neurosurgeon, there are many details to be discussed, so the discussion may be longer than average. And thank you for your suggestion, we have deleted some unnecessary sentences.

Reviewer 2 Report

I enjoyed this paper and congratulate the authors on their results in a very interesting and difficult patient population.  I think this is a great paper, I do a lot of cushing disease and this gives me a lot of guidance on reoperation and I would welcome this to the literature.  Great job.  

Author Response

Thank you very much for your positive comments.

Reviewer 3 Report

i have 2  major concerns:

1. the topic is not new and there are better studies than this one

2. there are not enough relevant  outcome parameters tested to support the supposed strategy

Author Response

Thank you for suggestions. As we all known, Cushing’s Disease is a very rare disease, an estimated incidence of 0.7-2.4 cases per million per year*, most of neurosurgical centers don’t have enough cases for analysis, and still has many problems to be solved, so may be the topic is not new, and in our opinion, it is worthy to be discussed again. In the last twenty years, there were 42 patients with persistent or recurring hypercortisolism in our center, it was not a small data for Cushing’s Disease, and we had analyzed the closely relevant outcome parameters.

* Rosario Pivonello, Andrea M Isidori, Maria Cristina De Martino, et al. Complications of Cushing's syndrome: state of the art . Lancet Diabetes Endocrinol. 2016 Jul;4(7):611-29. doi: 10.1016/S2213-8587(16)00086-3. Epub 2016 May 10.

Reviewer 4 Report

This is a very good review study of repeat transphenoidal surgery by one surgeon for recurrent/persistent Cushing's Disease (CD) after Transphenoidal surgery. It underlines the challenges of CD treatment, need for an experienced or well trained surgeon, comfortable with endoscopy and current technology, for better outcomes and less complications. It is well-done and well-written. The references should be listed alphabetically by first author for easier and systematic review.

Author Response

Thank you for your positive comments. Surely the transsphenoidal surgery for Cushing’s Disease should be done by an experienced or well-trained surgeon, especially for patients with persistent or recurring hypercortisolism after initial surgery. And the references were listed by the rules of JCM.

Round 2

Reviewer 3 Report

none

Author Response

Thank you for your suggestions. The English language has been polished by Cactus Communications (Shanghai) before the authors submitted the manuscript. We added complications of second TSS for permanent or recurrent CD, and the hormone replacement therapy, then the readers could weight the the pros and cons for the surgical strategies.